# Effectiveness of High-Pressure Processing Treatment for Inactivation of *Listeria monocytogenes* in Cold-Smoked and Warm-Smoked Rainbow Trout

**Kati Riekkinen [1],\*, Kati Martikainen [2] and Jenni Korhonen [1]**

[1] Institute of Public Health and Clinical Nutrition, University of Eastern Finland, 70211 Kuopio, Finland
[2] Department of Environmental and Biological Sciences, University of Eastern Finland, 70211 Kuopio, Finland
\* Correspondence: kati.riekkinen@uef.fi

**Abstract:** High-pressure processing (HPP) is a promising method for preserving food, for example by inactivating pathogens and spoilage microbes. However, there is still a lack of knowledge about the optimal processing parameters for different food products. The aim of this study was to assess the effectiveness of different pressures to inactivate *Listeria monocytogenes* in cold-smoked and warm-smoked rainbow trout (*Oncorhynchus mykiss*) ready-to-eat (RTE) fish products. *L. monocytogenes* ATCC 7644 was inoculated into sliced cold-smoked rainbow trout fillets and whole warm-smoked rainbow trout fillets. The fish samples were pressure-treated at three different pressures, either at 200, 400, or 600 MPa, at $4 \pm 1\,^\circ$C for 3 min in each pressure. Bacterial enumeration of the samples and control samples were analysed 1, 14, and 28 days after the HPP treatment by using serial dilution and the spread plate technique. Based on the study results, the most effective pressure was 600 MPa and the number of *L. monocytogenes* colonies, both in cold-smoked and in warm-smoked fish samples, was within the official limit of the Regulation (EC) No. 2073/2005 (100 cfu/g) even after 28 days of storage.

**Keywords:** high-pressure processing; *Listeria monocytogenes*; rainbow trout; *Oncorhynchus mykiss*; ready-to-eat

## 1. Introduction

The high-pressure processing (HPP) technique for inactivating spoilage and pathogenic bacteria was already introduced during the 19th century [1], but industrial applications were not launched until 1990 [2]. Thus, HPP is not a new method for processing and preserving food products. Additionally, the value of HPP has recently increased in response to consumers' increased interest in minimal processed food with healthy nutritional components. HPP is a nonthermal pasteurisation method for effectively inactivating vegetative pathogenic and spoilage microbes by hydrostatic pressure [3]. The pressure range in the process is between 100–1000 MPa depending on the food product [4,5]. The challenge of HPP is to ensure food microbiological safety for each individual product. The efficiency of HPP treatment depends on several factors, such as properties of the food matrix, parameters of the treatment, and microbe to be inactivated [3]. Thus, HPP must be customised based on several factors.

The main extrinsic factors influencing microbial inactivation in food products are the target pressure and the holding time [3]. Usually, a higher pressure and longer holding time increase inactivation of microbes [5,6]. Certain parameters, such as a pressure between 400–600 MPa for 2–7 min at room temperature, have been observed to be the most effective at decreasing vegetative spoilage microbes and foodborne pathogens, in some cases four logarithmic times or more [4]. According to the European Food Safety Authority (EFSA) Panel on Biological Hazards [3], a disadvantage is the absence of an indicator to ensure that each HPP treatment is sufficiently effective under factory conditions. Therefore, further

studies are required on the effectiveness of HPP to inactivate, e.g., different amounts of *Listeria* in ready-to-eat (RTE) fish products to achieve general minimum processing parameters, thereby ensuring microbiological safety [3].

*Listeria* bacteria are destroyed at 65 °C or higher, but can also contaminate food after heating. Fish and fish products, meats, cheeses, and raw vegetables have been reported to be the most likely foods in which *Listeria* is present [7,8]. For example, in 2020 listeriosis was the fifth most common zoonosis in humans in the EU, and 1876 *Listeria monocytogenes* invasive human cases were confirmed [8]. In the same year, *L. monocytogenes* was found in 4.3% of RTE fish and 4.1% of RTE fishery products in the EU area [8]. According to a study by Beaufort et al. [9], *L. monocytogenes* contamination was detected in 6.5% of the cold-smoked salmon samples after a storage time trial. Based on a study by Aalto-Araneda et al. [10], the main source of *Listeria* contamination is slicing or skinning machines in the processing environment and they found *Listeria* only in sliced RTE fish products. In addition, Lappi et al., [11] observed that most (80%) of the naturally *Listeria*-contaminated RTE cold-smoked salmon samples were sliced and vacuum-packed.

In a study by Erkan et al. [12], cold-smoked salmon samples were high-pressure treated. Based on sensory, chemical, and microbiological analysis, pressure treatment at 250 MPa at 3 °C for 5 min and at 250 MPa at 25 °C for 10 min increased the shelf life of the salmon samples by two weeks compared to nontreated control samples [12]. The pressure level of 600 MPa for 5 min effectively lowered the amount of *L. monocytogenes* (6.46 $\log_{10}$ cfu/g, $p < 0.01$) in mild-smoked rainbow trout fillets [13]. In addition, the treatment at 600 MPa for 1 min had a significant ($p < 0.01$) decreasing effect on the amount of *L. monocytogenes* in rainbow trout samples compared to nontreated control samples [13]. Despite the 600 MPa for 5 min treatment, not all *L. monocytogenes* bacteria in samples were inactivated, and the amount of *Listeria* increased significantly during the storage time [13]. In addition, after 26–41 days of storage, a low level (0.3–20 cfu/g) of *Listeria innocua* was observed in cold-smoked salmon treated at 900 MPa, although no *L. innocua* was detected after five days [14]. However, *L. innocua* was observed to be inactivated significantly ($p < 0.05$) even after 60 s at 500 MPa [14].

Lakshmanan and Dalgaard [15] observed that HPP treatment at 250 MPa for 20 min at a temperature of 9 °C did not inactivate *L. monocytogenes* in cold-smoked salmon samples but increased the lag-phase time of *Listeria*. The lag-phase time of *L. innocua* in cold-smoked salmon samples was also extended by HPP treatment above 600 MPa [14]. The extension of microbe lag-phase time also contributes to a longer shelf life of a food product even if the microbe is not totally destroyed. According to Basaran-Akgul et al. [16], HPP treatment at a pressure over 414 MPa for 5 min reduced tested *Listeria* strains by more than four logarithmic times in minced rainbow trout. In addition, sodium chloride (NaCl, 1% and 3%) enhanced the reduction of *Listeria* at higher pressure treatments (ca 500 MPa) [16]. A synergistic antimicrobial effect of high hydrostatic pressure (HHP) treatment and liquid smoke was also shown to reduce different *Listeria* strains, whereas NaCl had no significant effect for most of the samples [17]. However, the antimicrobial effect of NaCl also depends on the target microbe, as different *Listeria* strains react differently to NaCl [17]. Montero et al. [18] observed that smoke (phenol concentration 82 ppm) and NaCl (2.93%), together with high-pressure treatment (300 MPa for 15 min), prevented the growth of *L. monocytogenes* effectively in cold-smoked dolphinfish fillets during the storage time.

RTE food products are not heated before consumption, causing a risk of *Listeria* infection if raw material has already been contaminated or the final product has been contaminated during manufacturing. Moreover, cold-smoked fish products are prepared within a temperature range that is below the inactivation temperature of pathogens such as *Listeria*. Thus, HPP treatment could enhance the microbial safety and extend the shelf life of RTE products. The aim of this study was to assess the effectiveness of different HPP treatment pressures, 200, 400, and 600 MPa, on the inactivation of *L. monocytogenes* in cold-smoked and warm-smoked rainbow trout RTE fish products.

## 2. Materials and Methods

Sliced cold-smoked rainbow trout (*Oncorhynchus mykiss*) fillets and whole warm-smoked rainbow trout (*Oncorhynchus mykiss*) fillets were obtained from a local fish factory (Escamar Seafood Ltd., Kuopio, Finland). The fish products had been salted two days and smoked one day before vacuum-packing in the factory. Cold-smoked rainbow trout fillets were salted by the injection salting method, containing a salt solution and lactate solution (E326, E262). Warm-smoked rainbow trout fillets were also salted by the injection salting, containing only salt solution. The final NaCl content in the cold-smoked rainbow trout product was 2.5%, and in the warm-smoked rainbow trout product, 1.8%. Cold-smoked rainbow trout fillets were smoked 11 h and warm-smoked rainbow trout fillets were smoked 3 h with natural alder wood smoke. The smoking processes were performed using the parameters specified by the factory.

On the analysis day, the fish samples were cut with sterile knives to a weight of $25.0 \pm 0.5$ g and repacked in vacuum plastic bags. Before HPP treatment, an *L. monocytogenes* suspension was inoculated into the samples. In addition, control samples without added *L. monocytogenes* and without HPP treatment were analysed. All samples were stored at $4 \pm 2\,^{\circ}\text{C}$, also during the cutting and weighing.

The *Listeria monocytogenes* ATCC 7644 strain was cultured aerobically on Tryptone Soy (TS) agar plates (LabM, Lancashire, Greater Manchester, UK). For the test, a colony was transferred from a plate to the Tryptone Soy (TS) broth solution (LabM, Lancashire, Greater Manchester, UK) and cultured at $37\,^{\circ}\text{C}$ overnight. On the next day the bacterial culture was pelleted, washed, and suspended in physiological saline (0.9% NaCl). The turbidity of the microbial suspension was measured with a spectrophotometer (UV-1600PC, VWR, Leuven, Belgium) at a wavelength of 625 nm and the absorbance was set to 0.1–0.15. The suspension was further diluted 100-fold in a dilution solution (0.9% NaCl, 0.1% peptone). The microbial suspension concentration used in the test was $1 \times 10^6$ cfu/mL and 250 µL was pipetted into the fish samples. Inoculated samples were kept chilled ($4 \pm 2\,^{\circ}\text{C}$) until the HPP treatment on the next day.

The vacuum-packed fish samples were processed by HPP in a local food factory (Toripiha Ltd., Suonenjoki, Finland). The treatments were carried out in a hydrostatic press (Uhde-350-60, ThyssenKrupp Ag, Essen, Germany), using water as the pressurising fluid. The samples were pressure-treated at three different pressures, either at 200, 400, or 600 MPa, at $4 \pm 1\,^{\circ}\text{C}$ for 3 min at each pressure. All treatments of samples were repeated three separate times on different days.

Bacterial enumerations of the inoculated samples and control samples were analysed 1, 14, and 28 days after the HPP treatment. The sample was transferred into a sterile Stomacher bag and 225 mL of dilution solution (0.9% NaCl, 0.1% peptone) was added. The sample was stomached for 60 s at the speed 'normal' (Laboratory Blender Stomacher 400, Seward, GWB, England) and 10-fold serially diluted. Next, samples (100 µL) were placed onto plates of *Listeria*-selective agar, acc. Ottaviani and Agosti (ISO) (VWR Chemicals BDH, Leuven, Belgium), in duplicate using the spread plate technique. The plates were incubated for 48 h–72 h at $37\,^{\circ}\text{C}$ and the colonies were counted. The detection limit of the method was 100 cfu/g, which is also the official limit of *L. monocytogenes* according to Regulation (EC) No. 2073/2005 on commercial RTE food products, stating a product must not exceed the limit 100 cfu/g during its shelf life [19].

The data were not normally distributed, thus a nonparametric Kruskall-Wallis H test with Bonferroni correction was applied, and *p*-values less than 0.05 were considered statistically significant. Statistical analysis was performed by IBM SPSS Statistics version 27 software. The results lower than the detection limit were analysed as half (50 cfu/g) of the detection limit.

## 3. Results and Discussion

The results of the HPP treatments are presented in Tables 1 and 2. The numbers of *Listeria monocytogenes* colonies in inoculated samples, both in cold-smoked and in warm-smoked

fish samples, were within the official limit of the Regulation (EC) No. 2073/2005 (100 cfu/g) after the pressure treatment at 600 MPa in every analysed timepoint. A significant difference ($p < 0.05$) was observed between inoculated nontreated cold-smoked fish samples and inoculated cold-smoked fish samples treated at 600 MPa at the one-day analysis point. In addition, a significant difference ($p < 0.05$) between inoculated nontreated warm-smoked fish samples and inoculated warm-smoked fish samples treated at 600 MPa was observed at the 28-day analysis point. *L. monocytogenes* was not detected in control samples with no added *L. monocytogenes* and no HPP treatment (the results are not presented in the tables), indicating the good microbial quality of the raw material used in this study.

**Table 1.** Numbers ($\log_{10}$ cfu/g) of *Listeria monocytogenes* colonies in inoculated cold-smoked rainbow trout samples at three (1, 14, and 28 days) different storage timepoints.

| Treatment/Storage Time | 1 Day | 14 Days | 28 Days |
|---|---|---|---|
| Nontreated | 5.5 ± 0.1 [a] | 5.5 ± 0.7 | 4.2 ± 1.5 |
| HPP treated at 200 MPa | 5.4 ± 0.2 | 5.2 ± 0.7 | 3.7 ± 0.8 |
| HPP treated at 400 MPa | 4.6 ± 0.4 | 4.4 ± 0.5 | 2.5 ± 0.9 |
| HPP treated at 600 MPa | [1] [a] | [1] | [1] |

Values are the mean ± standard deviations of three replicates. Values followed by the same letter are significantly different ($p < 0.05$). [1] Below the detection limit (100 cfu/g).

**Table 2.** Numbers ($\log_{10}$ cfu/g) of *Listeria monocytogenes* colonies in inoculated warm-smoked rainbow trout samples at three (1, 14, and 28 days) different storage timepoints.

| Treatment/Storage Time | 1 Day | 14 Days | 28 Days |
|---|---|---|---|
| Nontreated | 4.1 ± 0.1 | 5.3 ± 1.2 | 6.5 ± 1.1 [a] |
| HPP treated at 200 MPa | 3.9 ± 0.2 | 5.0 ± 0.8 | 5.3 ± 1.5 |
| HPP treated at 400 MPa | 2.9 ± 0.4 | 3.5 ± 0.9 | 3.9 ± 1.7 |
| HPP treated at 600 MPa | [1] | [1] | 2.0 ± 0.5 [a] |

Values are the mean ± standard deviations of three replicates. Values followed by the same letter are significantly different ($p < 0.05$). [1] Below the detection limit (100 cfu/g).

The amount of *L. monocytogenes* was observed to decrease during the storage time in inoculated cold-smoked rainbow trout samples in both nontreated and treated samples. In a study by Basaran-Akgul et al. [16], HPP treatment at a pressure over 414 MPa for 5 min reduced the amount of *Listeria* by more than four logarithmic times in minced rainbow trout samples, and NaCl was found to further increase the effectiveness of the treatment. In this study, the reduction of the amount of *L. monocytogenes* was the highest (3.8-log) in the cold-smoked fish samples at the one-day analysis point for the treatment at 600 MPa.

However, the amount of *L. monocytogenes* increased during the storage time in all inoculated warm-smoked fish samples, although the growth was weaker in samples with higher-pressure treatments. Thus, the regrowth of *Listeria* was observed in all (nontreated and treated) warm-smoked fish samples during storage time. The HPP treatment at 600 MPa for 3 min is not sufficient to destroy *L. monocytogenes* in warm-smoked rainbow trout samples if the fish material is heavily contaminated with *Listeria*. Similar findings were detected in a study by Mengden et al. [13], as after 41 storage days the amount of *Listeria* in HPP-treated mild-smoked rainbow trout samples reached almost the same level as the *Listeria* amount in non-HPP-treated samples. These results clearly indicate that *Listeria* can grow effectively at refrigerator temperatures.

Sodium chloride (NaCl) is one of the main ingredients in RTE fish products. NaCl is the factor that gives the products their characteristic flavour, taste, colour, and texture. For example, NaCl, liquid smoke and drying, vacuum packing, and low storage temperature were able to control *L. monocytogenes* in cold-smoked salmon samples [20]. Thus, it seems that a high NaCl concentration can effectively decrease the amount of *L. monocytogenes* in cold-smoked fish samples, even in the non-HPP-treated samples. Furthermore, smoke extract could be an additional factor that could be used to inhibit the growth of *Listeria* in

vacuum-packed cold-smoked rainbow trout [21]. Smoking decreases the pH of the fish [18], and low pH (4.5) was observed to enhance the reduction of *L. monocytogenes* at the 200 MPa HPP treatment compared to higher pH [22].

Overall, a good microbiological quality of fresh rainbow trout fillets was reached by HPP treatment at 450 MPa and 600 MPa for 15 min as the microbial growth of total aerobic bacteria was not observed after 6 days of storage at 4 °C [23]. In the present study, an acceptable level of *L. monocytogenes* was reached after the treatment at 600 MPa for 3 min for smoked RTE fish products. However, Gudbjornsdottir et al. [14], indicated in their study that high pressure combined with short holding time is the most effective way to inactivate *L. innocua* in cold-smoked salmon, and the microbial safety of a fish product can be ensured using the treatment at 700–900 MPa for 10 s.

The inactivation of *L. monocytogenes* was observed to increase with higher pressure and longer holding time [13]. In our study, we did not test the influence of different holding times, but higher-pressure treatment increased inactivation of *L. monocytogenes* in both cold-smoked and warm-smoked rainbow trout samples. Based on the results by Ekonomou et al. [17], HHP treatment at the low-pressure level, together with a low amount of liquid smoke, could be a promising method for decreasing *Listeria*. For example, the amount of nonresistant *Listeria* strains can be reduced more than four logarithmic times of cfu/mL with the combined effect of the HHP treatment and liquid smoke [17].

Salmon fish species have good nutritional fat compositions, including a high amount of unsaturated fatty acids and especially n-3 fatty acids. Yagiz et al. [24] observed that HPP is a mild treatment for fatty acids in salmon samples, as it does not change the n-3 and n-6 polyunsaturated fatty acid profile. Furthermore, the treatment at 300 MPa for 15 min decreased susceptibility to oxidation of the samples compared to control samples, thus improving quality and extending shelf-life during the storage time [24]. However, the structure of proteins can change during the pressure treatment, for example causing denaturation, aggregation, or gelation of proteins [25]. Thus, HPP treatment may also lead to undesirable changes in protein structure while changing the sensorial and texture properties of the product. Cioca et al. [26] observed that significant protein denaturation in rainbow trout samples occurred at pressures of 400–600 MPa at holding times of 3 or 6 min, negatively affecting protein structure. However, appropriate specific parameters (pressure, time, temperature) of the treatment could extend the shelf life of the product without changing the sensorial properties, but the parameters should be optimised on a case-by-case basis.

## 4. Conclusions

Nonthermal food preserving technologies, such as HPP, are emerging in Europe. Therefore, EFSA also requires more results based on food studies using HPP technology to ensure product safety [3]. This study demonstrated that HPP treatment can improve the microbial safety of smoked RTE fish products and prevent the growth of pathogenic *L. monocytogenes* in cold-smoked and warm-smoked rainbow trout. Use of natural antimicrobials, such as smoke with a moderate salt concentration, could also increase the efficacy of the HPP treatment. However, to ensure an efficient treatment, the pressure must be at least 600 MPa for 3 min when a high bacterial exposure dose is applied. When applying the treatment of 600 MPa, despite a high amount of inoculated *Listeria*, after 28 storage days the numbers of *L. monocytogenes* colonies, both in cold-smoked and in warm-smoked RTE fish samples, were within the official limit of the Regulation (EC) No. 2073/2005.

Finally, by using suitable processing parameters, HPP could improve microbiological safety and increase the shelf life of cold-smoked and warm-smoked RTE fish products without chemical preservatives or heating. However, the utmost important factors are good microbiological quality of raw materials followed by good manufacturing practices, which together can ensure a high-quality product during the whole shelf life. In addition, fish products require unbroken low-temperature storage during the whole supply chain.

**Author Contributions:** Conceptualization, J.K.; methodology, J.K., K.R. and K.M.; validation, K.R. and K.M.; formal analysis, K.M. and K.R.; investigation, K.R. and K.M.; resources, J.K.; data curation, J.K.; writing—original draft preparation, K.R.; writing—review and editing, K.R., K.M. and J.K.; visualization, K.R. and K.M.; supervision, J.K.; project administration, J.K.; funding acquisition, J.K. All authors have read and agreed to the published version of the manuscript.

**Funding:** This work was funded by the EU European Regional Development Fund and the Regional Council of Northern Savo "Improving Shelf-Life and Ensuring Quality in Food Using New Technologies" development project 2019–2022.

**Institutional Review Board Statement:** Not applicable.

**Informed Consent Statement:** Not applicable.

**Data Availability Statement:** Not applicable.

**Acknowledgments:** We wish to thank Tiina Tiussa from Escamar Finland Ltd., Kuopio, for providing the cold and warm smoked rainbow trout for our study and Juho Kylmälä from Toripiha Ltd., Suonenjoki, Finland, for sharing his expertise on high pressure processing.

**Conflicts of Interest:** The authors declare no conflict of interest. The funders had no role in the design of the study; in the collection, analyses, or interpretation of data; in the writing of the manuscript; or in the decision to publish the results.

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
