# Peer review of "Effectiveness of High-Pressure Processing Treatment for Inactivation of Listeria monocytogenes in Cold-Smoked and Warm-Smoked Rainbow Trout"

_applsci, doi:10.3390/app13063735_

Round 1

Reviewer 1 Report

The manuscript entitled (Effectiveness of high-pressure processing (HPP) treatment for inactivation of Listeria monocytogenes in cold-smoked and  warm-smoked rainbow trout) the authors find a very interesting results by using HPP treatment as food preservation method.

Some suggestions as following:

1- Avoid using abbreviation in the title. So delete HPP from the title.

2-  I suggest to delete HPP from the keyword.

3-  I suggest to add ready-to-eat in the keyword.

4- Can we apply more than 600 MPa, may be you will get more interesting results?

5- Why it is only 3 min of Appling HPP? What will happened if the time become more than 3 min in this type of the treated foods?

6- I understand that the study is focus on the microbial aspect, put it is important to  add more results if you have about the eating quality of cold-smoked and warm smoked rainbow trout.

Reviewer 2 Report

The article is clearly written. The experimental design is straightforward. My only comment is that it would be better to use a cocktail culture rather than a single strain to test the efficacy of HPP to minimize strain variance on the resistance or sensitivity of HPP. 

Reviewer 3 Report

The central idea of the research is not exactly new, there are already many studies on the effect of high-pressure treatment on fish fillets, cold and hot smoked, so I don't know how necessary it would be to document this specific aquaculture species.

On the other hand, it is a single analysis; the authors only report their results with a couple of tables, which seems insufficient to have a paper's relevance. Moreover, the authors must be aware that there are other fourth technology technologies, such as massive sequencing, among others, which could enrich this research.

I thought about finding at least predictive or response surface graphs in this manuscript that could give more information about the processing technology. 

Round 2

Reviewer 3 Report

No more coments